# The relationship between emotional disorders and heart rate variability: A Mendelian randomization study

**Xu Luo[1], Rui Wang[1], YunXiang Zhou[1], Wen Xie [1,2]***

**1** College of Clinical Medicine, University of Traditional Chinese Medicine, Chengdu, Sichuan, China,
**2** Department of Cardiology, Hospital of Chengdu University of Traditional Chinese Medicine, Chengdu, Sichuan, China

* 18428356338@163.com

## Abstract

### Objective

Previous studies have shown that emotional disorders are negatively associated with heart rate variability (HRV), but the potential causal relationship between genetic susceptibility to emotional disorders and HRV remains unclear. We aimed to perform a Mendelian randomization (MR) study to investigate the potential association between emotional disorders and HRV.

### Methods

The data used for this study were obtained from publicly available genome-wide association study datasets. Five models, including the inverse variance weighted model (IVW), the weighted median estimation model (WME), the weighted model-based method (WM), the simple model (SM) and the MR–Egger regression model (MER), were utilized for MR. The leave-one-out sensitivity test, MR pleiotropy residual sum and outlier test (MR-PRESSO) and Cochran's Q test were used to confirm heterogeneity and pleiotropy.

### Results

MR analysis revealed that genetic susceptibility to broad depression was negatively correlated with HRV (pvRSA/HF) (OR = 0.380, 95% CI 0.146–0.992; p = 0.048). However, genetic susceptibility to irritability was positively correlated with HRV (pvRSA/HF, SDNN) (OR = 2.017, 95% CI 1.152–3.534, p = 0.008) (OR = 1.154, 95% CI 1.000–1.331, p = 0.044). Genetic susceptibility to anxiety was positively correlated with HRV (RMSSD) (OR = 2.106, 95% CI 1.032–4.299; p = 0.041). No significant directional pleiotropy or heterogeneity was detected. The accuracy and robustness of these findings were confirmed through a sensitivity analysis.

### Conclusions

Our MR study provides genetic support for the causal effects of broad depression, irritable mood, and anxiety on HRV.

**Data Availability Statement:** All relevant datas are within the manuscript and its Supporting Information files.

**Funding:** This research was financially supported by the Key Project of Sichuan Provincial

Department of Science and Technology'funding
(NO. 2022YFS0395). State Administration of
Traditional Chinese Medicine (NO. 2023MS522.

## 1. Introduction

Emotions are continually evoked by stimuli from internal and external environments, and emotions help humans establish connections with each other [1]. The predominance of emotional disorders is mainly caused by difficulty in autonomously regulating or adapting to emotions. Emotional disorders are intrinsic characteristics of personality disorders [2], and they are coexisting symptoms of mental illness. Emotional disorders are characterized by changes in mood or emotions. Furthermore, they are closely associated with many neurological diseases and can impact the emotional, physical, and cognitive levels of patients [3]. The latest report from the Global Burden of Disease Research indicates an increase in the prevalence and burden of emotional disorders over the past two decades [4]. Additionally, emotional disorders have become more common since the COVID-19 epidemic [5], resulting in a high socioeconomic burden. Early recognition of emotional disorders can reduce the incidence and mortality of related diseases [6].

Heart rate variability (HRV) refers to the alteration between two successive heartbeats and is generally measured using time-domain metrics, such as the standard deviation of the normal-to-normal interbeat intervals (SDNN), the root mean square of the successive differences in interbeat intervals (RMSSD), and peak-valley respiratory sinus arrhythmia or high frequency power (pvRSA/HF) [4, 7]. Detecting HRV is not only crucial in evaluating cardiovascular health but also in assessing the parasympathetic and sympathetic components of the autonomic nervous system [8, 9]. Recent studies have demonstrated that HRV analysis also plays an essential role in emotional research and detection [10]. It can be a valuable tool for tracking changes in emotional state and detecting early onset and relapse in individuals with emotional disorders [11]. Previous studies have shown that individuals with emotional disorders are characterized by chronic low heart rate variability compared to healthy individuals [12, 13], but it remains unclear whether a causal relationship exists between emotional disorders and HRV.

Mendelian randomization (MR) analysis uses germline genetic variants as exposure instrumental variables (IVs) to study the causal relationships between the exposure phenotype and the outcome phenotype [14]. Because exposure instrumental variables (IVs) during pregnancy are randomly allocated, they are unlikely to be affected by disease status. Therefore, MR studies can exclude unobserved confounding factors as well as reverse causal relationships and overcome the typical pitfalls present in observational studies [15, 16]. In the present study, we aimed to use MR analysis to investigate the genetic relationships between seven emotional disorders (broad depression, major depressive disorder (MDD), obsessive-compulsive disorder (OCD), bipolar disorder, irritable mood, anxiety, and mania) and HRV.

## 2. Materials and methods

### 2.1 Study design and data sources

We conducted MR analysis to assess evidence for causal effects of emotional disorders on HRV. An overview of the research design is presented in Fig 1. To obtain valid results, MRs need to fulfill three key assumptions: (a) genetic variants used in the analysis should be significantly associated with the exposure; (b) genetic variants extracted as instrumental variables for exposure should be independent of confounding factors that are associated with the selected exposure and outcome; and (c) genetic variants should affect the outcome only through exposure and not via other biological pathways [17].

We explored publicly available GWAS databases to access eligible datasets on exposure and outcomes. These included databases such as the GWAS catalog, the IEU open GWAS, and the

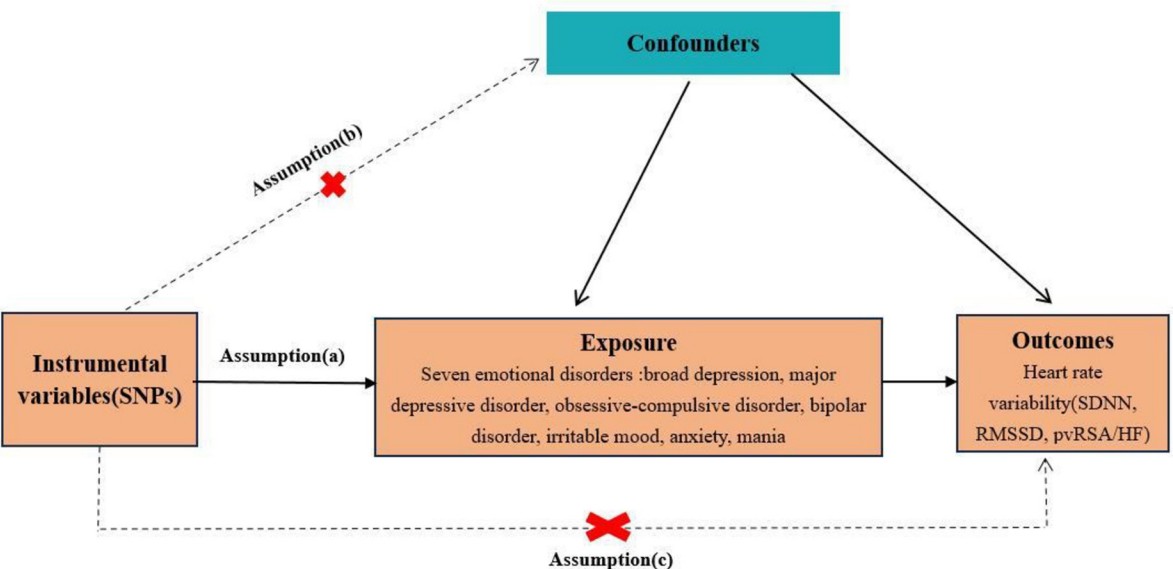

**Fig 1. Schematic representation of the Mendelian randomization analysis.** The continuous lines represent the relationships that hold in MR analysis. Broken lines represent associations that violate the assumptions of Mendelian randomization.

Phe Web. Thus, no further ethical approval was necessary. Since mixed populations may lead to partial estimates, we restricted the genetic background of individuals included in the MR study to those of European ancestry [18].

The data for broad depression originated from the UK Biobank's GWAS (GWAS ID: ebi-a-GCST005902) [19]. This dataset included 322,580 Europeans (113769 cases and 208811 controls), and 7,624,934 single nucleotide polymorphisms (SNPs) were identified. The data for major depressive disorder (MDD) were retrieved from the NA genome-wide genetic environment analysis (GWAS ID: ebi-a-GCST009979) [20]. The dataset included 92,957 Europeans (29,475 cases and 63,482 controls), and 7,767,934 SNPs were identified. The data on obsessive-compulsive disorder were derived from the PGC genome-wide association analysis (GWAS ID: ieu-a-1189). The dataset comprised 33,925 Europeans (26,888 cases and 7,037 controls), and 8,409,517 SNPs were identified. The data for patients with bipolar disorder were acquired from the GWAS (GWAS ID: ieu-b-41) conducted by the Bipolar Disorder Working Group of the Psychiatric Genomics Consortium [21]. The dataset included 51710 Europeans (20352 cases and 31358 controls), and 13413244 single nucleotide polymorphisms (SNPs) were identified.

The data on irritable mood (GCST006941) were obtained from Nagel M's genome-wide genotyping array [22]. The dataset included 366,726 Europeans, and 10,847,151 SNPs were identified. The data on anxiety disorders (GCST90267276) were obtained from the Genome-wide genotyping array conducted by Schoeler T [23], which analyzed 282802 Europeans and identified 7296966 SNPs. The data on mania (GCST90044038) were obtained from a genome-wide genotyping array that was carried out by Jiang L [24], which analyzed 146837 Europeans (8,449 cases and 138,388 controls) and identified 11842647 SNPs.

The data presented in the results section were derived from the GWAS public database [25]. Three indices for HRV were used: one dataset (pvRSA/HF) (GWAS ID: ebi-a-GCST004734) included 24088 Europeans and identified 2526821 SNPs; one dataset (RMSSD) (GWAS ID: ebi-a-GCST004733) included 26523 Europeans and identified 2528639 single nucleotide polymorphisms (SNPs); and one dataset (SDNN) (GWAS ID: ebi-a-GCST004734) included 27850 Europeans and identified 2549728 SNPs.

## 2.2 Selection of instrumental variables

To include eligible instrumental variables, all SNPs in strong linkage disequilibrium (LD) ($r^2 = 0.001$ and kb = 10000) were excluded. The screening threshold was set at $p<5 \times 10-8$, but some exposure data cannot meet the minimum requirements for MR studies with 10 qualified instrumental variables; therefore, a less stringent threshold of $p<5 \times 10-6$ was used [26]. Considering the relatively loose threshold, to reduce potential weak instrumental bias, we calculated the F-statistic. $F = R^2(N\text{-}2)/(1\text{-}R^2)$ Within them, $R^2$ indicates the extent to which the instrumental variable explains the factor of exposure. $R^2 = 2*(1 - MAF)*MAF*\frac{\beta}{SD}$ [27] (MAF denotes the minor allele frequency, β is the effect size for the genetic variant of interest, and SD stands for standard deviation). Given that EAF (effect allele frequency) data could not be obtained for some samples, these data were taken $F = Beta^2/Se^2$ (beta (β) and Se data can be directly obtained, where beta (β) is the effect size for the genetic variant of interest and Se denotes the standard error value of beta (β)) [28, 29]. Instrumental variables with F>10 were included in the MR analysis [30]. After intersection of the exposure and outcome data, palindromic SNPs with moderate allele frequencies were removed. Finally, we identified eight SNPs that are broadly associated with depression. Deletion of two palindromic SNPs, namely, rs10947863 and rs7568777, and six SNPs were associated with major depressive disorder ($p<5 \times 10-6$). Four OCD-related SNPs ($p<5 \times 10-8$) were identified after deletion of the palindromic SNP rs909701. After deletion of three palindromic SNPs, namely, rs13231398, rs2314398 and rs5758065, five SNPs associated with bipolar disorder were obtained ($p<5 \times 10-8$). Five palindromic SNPs, namely, rs10228350, rs17151565, rs343949, rs4781534, and rs9630740, were deleted to obtain 16 SNPs associated with irritable mood ($p<5 \times 10-8$). One palindromic SNP, namely, rs4073568, was deleted to obtain 13 SNPs associated with anxiety disorder ($p<5 \times 10-6$) and 5 SNPs associated with mania ($p<5 \times 10-6$). These SNPs serve as appropriate instrumental variables for exploring the potential causal relationship between exposure and outcomes (S1 Table).

## 2.3 Statistical analysis

The MR-PRESSO software package (version 1.0.0), R software (version 4.3.1), TwoSampleMR (version 0.5.7), gwasglue (version 0.0.0.9000) and gwasvcf (version 0.1.1) packages were used to conduct dual-sample MR analysis.

This study used the inverse variance weighted model (IVM), weighted median estimation model (WME), weighted model-based methods (WM), simple model (SM) and MR Egger model (MER) to estimate the causal relationship between emotional disorders and heart rate variability. The classic IVW is used for the main MR analysis. This approach provides a consistent estimate of the causal effect of exposure on outcomes [31]. The WME is used to combine data from multiple genetic variants into a single causal estimator. Even when up to 50% of the information comes from invalid IVs, this estimator is consistent. It can reduce type 1 errors and estimate more precise causal relationships in the presence of horizontal pleiotropy [32]. Horizontal pleiotropy determines whether a genetic instrument affects an outcome through pathways other than via the exposure of interest [33]. The WM method can provide a robust causal effect estimate for horizontal pleiotropy [34]. The MER method can assess whether genetic variation has a nonzero mean pleiotropy effect on outcomes (directed pleiotropy), and under weaker assumptions, the modification of Egger regression (MR Egger) can be used to detect and correct bias caused by directed pleiotropy [35, 36]. We also used Q-statistics to measure whether there was heterogeneity between instruments. Heterogeneity is caused both by random sampling error and by inherent differences in research design and implementation methods, resulting in changes in observed research estimates [37]. When Q values were

significant ($P<0.05$), we used a multiplicative random effects model based on IVW. In all the other cases, a fixed effects model was used. Further application of MR Egger intercept data and the MR PRESSO method was used to test for horizontal pleiotropy and correct for horizontal pleiotropy by excluding outliers [38]. The stability of MR results is determined by excluding IVs one at a time in a sensitivity test [39].

## 3. Results

### 3.1 Emotional disorders and HRV (pvRSA/HF)

The results for the selected emotional disorders and HRV (pvRSA/HF) are shown in Table 1. Scatter plots are shown in S1 Fig. The IVM results showed that genetic susceptibility to broad depression may be associated with HRV (pvRSA/HF) (OR = 0.380, 95% CI = 0.146 0.992; p = 0.048). The WM results showed that genetic susceptibility to irritability may be associated with HRV (pvRSA/HF) (OR = 2.017, 95% CI (1.152 3.534), p = 0.008). There was almost no evidence for a causal relationship between other emotional disorders and HRV (measured by the pvRSA/HF). Moreover, Cochran's Q test showed no significant heterogeneity between emotional disorders and HRV (pvRSA/HF). In addition, the MR Egger intercept method and the MR PRESO method did not reveal a significant horizontal pleiotropic effect between these SNPs (S2 Table). According to the results of the leave-one-out analysis, after excluding one SNP at a time, the estimated risk of emotional disorders remained consistent with the risk of HRV (pvRSA/HF) (S2 Fig).

### 3.2 Emotional disorders and HRV (RMSSD)

The results for the selected emotional disorders and HRV (RMSSD) are shown in Table 2. Scatter plots are shown in S3 Fig. The IVM results showed that genetic susceptibility to anxiety may be associated with HRV (RMSSD) (OR = 2.106, 95% CI (1.032–4.299), p = 0.041). There is no evidence of a causal relationship between other emotional disorders and HRV (RMSSD). Moreover, Cochran's Q test showed no significant heterogeneity between emotional disorders and HRV (RMSSD). In addition, the MR Egger intercept method and the MR PRESO method

**Table 1. Mendelian randomization analysis results for emotional disorders and HRV (pvRSA/HF).**

| Exposure | Outcome | Inverse variance weighted | | MR Egger | | Weighted median | | Simple mode | | Weighted mode | |
|---|---|---|---|---|---|---|---|---|---|---|---|
| | | OR(95%CI) | P | OR(95%CI) | P | OR(95%CI) | P | OR(95%CI) | P | OR(95%CI) | P |
| Broad depression | | 0.380 (0.146–0.992) | 0.048 | 0.191 (0.001–25.009) | 0.531 | 0.703 (0.183–2.703) | 0.608 | 0.908 (0.121–6.779) | 0.927 | 0.940 (0.107–8.220) | 0.957 |
| Major Depressive Disorder | | 1.032 (0.864–1.232) | 0.731 | 0.079 (0.011–0.540) | 0.061 | 1.047 (0.874–1.254) | 0.616 | 1.169 (0.886–1.543) | 0.319 | 1.104 (0.851–1.433) | 0.490 |
| Obsessive Compulsive Disorder | Heart rate variability (pvRSA/HF) | 0.977 (0.898–1.063) | 0.589 | 0.818 (0.632–1.058) | 0.265 | 0.943 (0.872–1.019) | 0.139 | 0.939 (0.855–1.030) | 0.275 | 0.936 (0.855–1.026) | 0.252 |
| Bipolar Disorder | | 1.019 (0.888–1.170) | 0.789 | 1.101 (0.345–3.515) | 0.881 | 1.013 (0.872–1.178) | 0.864 | 0.980 (0.766–1.252) | 0.877 | 1.007 (0.802–1.265) | 0.953 |
| Irritable Mood | | 1.426 (0.892–2.280) | 0.138 | 2.302 (0.034–154.164) | 0.703 | 2.017 (1.152–3.534) | 0.008 | 2.241 (0.925–5.427) | 0.094 | 2.241 (0.934–5.375) | 0.091 |
| Anxiety Disorder | | 2.519 (0.574–11.051) | 0.221 | 0.537 (0.000–4732.949) | 0.896 | 2.149 (0.264–17.467) | 0.474 | 4.237 (0.133–134.804) | 0.429 | 3.735 (0.138–101.432) | 0.449 |
| Mania | | 0.990 (0.869–1.127) | 0.874 | 2.238 (0.528–9.488) | 0.354 | 0.980 (0.832–1.155) | 0.810 | 1.086 (0.873–1.352) | 0.501 | 0.933 (0.755–1.152) | 0.553 |

**Table 2. Results of the Mendelian randomization analysis for emotional disorders and HRV (RMSSD).**

| Exposure | Outcome | Inverse variance weighted | | MR Egger | | Weighted median | | Simple mode | | Weighted mode | |
|---|---|---|---|---|---|---|---|---|---|---|---|
| | | OR(95%CI) | P | OR(95%CI) | P | OR(95%CI) | P | OR(95%CI) | P | OR(95%CI) | P |
| Broad depression | | 1.028 (0.617–1.712) | 0.917 | 1.243(0.09–16.764) | 0.875 | 1.344 (0.712–2.539) | 0.362 | 1.547 (0.537–4.454) | 0.446 | 1.516 (0.548–4.193) | 0.449 |
| Major Depressive Disorder | | 1.034 (0.974–1.098) | 0.274 | 0.44(0.174–1.114) | 0.158 | 1.013 (0.937–1.094) | 0.754 | 1.001 (0.892–1.124) | 0.986 | 1.005 (0.903–1.119) | 0.929 |
| Obsessive Compulsive Disorder | Heart rate variability (RMSSD) | 0.995 (0.967–1.024) | 0.738 | 0.968(0.875–1.072) | 0.596 | 1.002 (0.969–1.036) | 0.915 | 1.003 (0.958–1.051) | 0.900 | 1.003 (0.956–1.053) | 0.904 |
| Bipolar Disorder | | 1.012 (0.956–1.07) | 0.690 | 1.06(0.701–1.603) | 0.801 | 1.004 (0.942–1.069) | 0.909 | 0.998 (0.911–1.094) | 0.974 | 0.999 (0.919–1.085) | 0.975 |
| Irritable Mood | | 1.217 (0.988–1.499) | 0.065 | 1.181 (0.181–7.699) | 0.864 | 1.243 (0.952–1.623) | 0.110 | 1.543 (0.899–2.649) | 0.136 | 1.494 (0.929–2.403) | 0.119 |
| Anxiety Disorder | | 2.106 (1.032–4.299) | 0.041 | 14.636 (0.212–1012.318) | 0.240 | 1.633 (0.654–4.077) | 0.293 | 1.684 (0.437–6.480) | 0.463 | 1.684 (0.456–6.222) | 0.450 |
| Mania | | 0.958 (0.899–1.021) | 0.186 | 0.914 (0.450–1.853) | 0.818 | 0.961 (0.895–1.032) | 0.277 | 0.966 (0.877–1.063) | 0.515 | 0.965 (0.885–1.053) | 0.469 |

did not reveal a significant horizontal pleiotropic effect between these SNPs (S3 Table). According to our leave-one-out analysis, after excluding one SNP at a time, the estimated effect of emotional disorders on the risk of HRV (RMSSD) did not change drastically (S4 Fig).

### 3.3 Emotional disorders and HRV (SDNN)

The results for the selected emotional disorders and HRV (SDNN) are shown in Table 3. Scatter plots are shown in S5 Fig. According to the IVM results, genetic susceptibility to irritable mood may be associated with HRV (SDNN) (OR = 1.154, 95% CI (1.000–1.331), p = 0.044); however, no evidence suggests such a relationship for other emotional disorders and HRV (SDNN). Additionally, Cochran's Q test revealed no significant heterogeneity between emotional disorders and HRV (SDNN). Furthermore, the MR Egger intercept method and the MR PRESO method did not reveal a notable horizontal pleiotropic effect between these SNPs

**Table 3. Results of the Mendelian randomization analysis for emotional disorders and HRV (SDNN).**

| Exposure | Outcome | Inverse variance weighted | | MR Egger | | Weighted median | | Simple mode | | Weighted mode | |
|---|---|---|---|---|---|---|---|---|---|---|---|
| | | OR(95%CI) | P | OR(95%CI) | P | OR(95%CI) | P | OR(95%CI) | P | OR(95%CI) | P |
| Broad depression | | 1.125 (0.802–1.578) | 0.496 | 1.612 (0.333–7.819) | 0.575 | 1.058 (0.690–1.623) | 0.796 | 0.930 (0.452–1.915) | 0.849 | 0.954 (0.478–1.901) | 0.896 |
| Major Depressive Disorder | | 1.021 (0.961–1.084) | 0.506 | 0.401 (0.203–0.793) | 0.058 | 0.971 (0.913–1.033) | 0.349 | 0.971 (0.871–1.081) | 0.613 | 0.969 (0.905–1.037) | 0.407 |
| Obsessive Compulsive Disorder | Heart rate variability (SDNN) | 1.008 (0.988–1.029) | 0.432 | 0.990 (0.923–1.061) | 0.799 | 1.015 (0.989–1.041) | 0.255 | 1.017 (0.982–1.053) | 0.417 | 1.017 (0.983–1.052) | 0.413 |
| Bipolar Disorder | | 0.985 (0.945–1.026) | 0.462 | 1.020 (0.758–1.372) | 0.905 | 0.979 (0.931–1.029) | 0.404 | 0.972 (0.905–1.043) | 0.473 | 0.975 (0.913–1.042) | 0.502 |
| Irritable Mood | | 1.154 (1.000–1.331) | 0.044 | 1.133 (0.320–4.014) | 0.849 | 1.191 (0.990–1.432) | 0.063 | 1.438 (0.952–2.173) | 0.105 | 1.413 (0.966–2.068) | 0.095 |
| Anxiety Disorder | | 1.473 (0.879–2.467) | 0.142 | 4.532 (0.262–78.415) | 0.321 | 1.178 (0.576–2.409) | 0.653 | 1.097 (0.331–3.640) | 0.882 | 0.982 (0.299–3.222) | 0.977 |
| Mania | | 0.982 (0.939–1.028) | 0.445 | 1.014 (0.607–1.694) | 0.961 | 0.978 (0.927–1.031) | 0.411 | 1.021 (0.946–1.102) | 0.616 | 0.963 (0.891–1.040) | 0.387 |

(S4 Table). During the leave-one-out analysis, we observed that the estimated risk of emotional disorders remained consistent with the risk of HRV (SDNN) even after excluding one SNP each time (S6 Fig).

The causal relationships between emotional disorders predicted by genes and the risk of heart rate variability and its subtypes are shown in Fig 2.

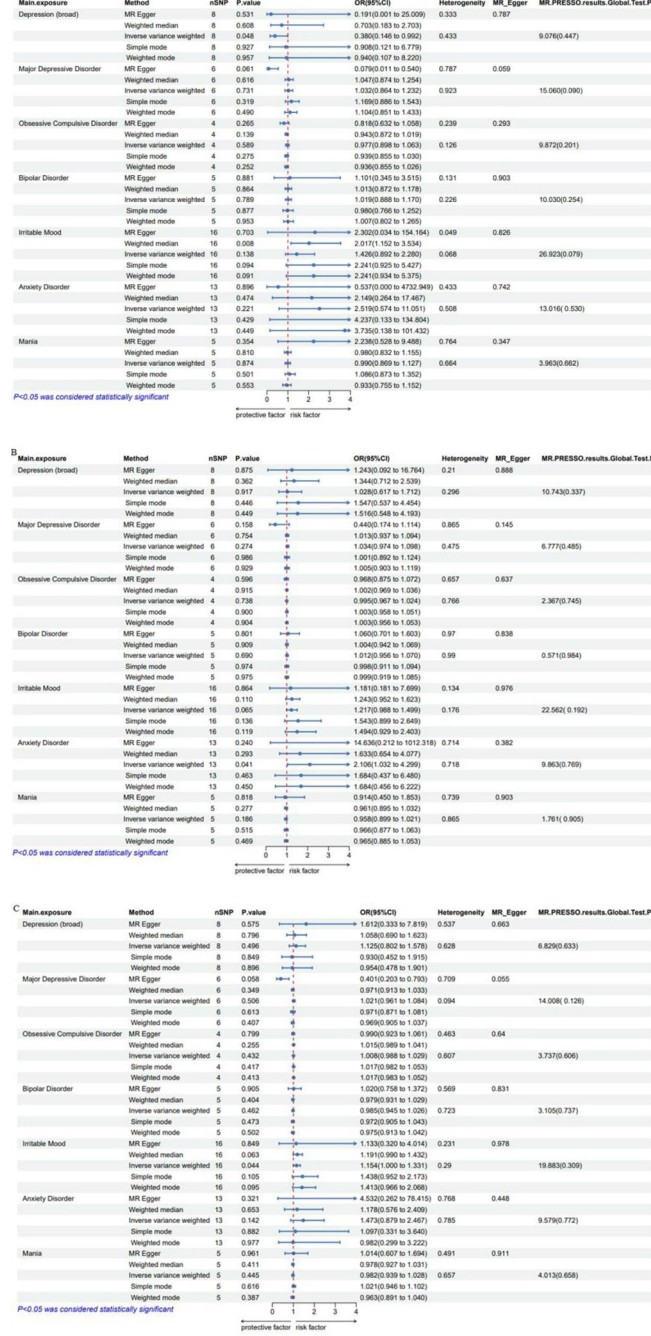

**Fig 2. Forest plot illustrating the causal relationships between gene-predicted emotional disorders heart rate variability, and the risk of its subtypes.** A: Emotional disorders and HRV (pvRSA/HF). B: Emotional disorders and HRV (RMSSD). C: emotional disorders and HRV (SDNN).

## 4. Discussion

This study examined the causal relationships between seven emotional disorders, namely, broad depression, severe depressive disorder, obsessive-compulsive disorder, bipolar disorder, irritable mood, anxiety, and manic disorder, and HRV (RMSSD, SDNN, and pvRSA/HF). The study concluded that broad depression is widely associated with a decrease in HRV (pvRSA/HF), irritable mood may be associated with an increase in HRV (SDNN, pvRSA/HF), and anxiety may be associated with an increase in HRV (RMSSD). The study revealed no causal relationships between severe depression, obsessive-compulsive disorder, bipolar disorder, or mania and HRV.

Numerous observational studies and reviews have discussed the link between emotional disorders and heart rate variability. Resting electrocardiograms were recorded for 15 minutes in both the depression patient group and the healthy control group, as documented by Ralf Hartmann and colleagues. The electrocardiogram data were preprocessed to extract heart beat intervals. Following the extraction of HRV parameters using both linear and nonlinear methods, an analysis of differences in these parameters at baseline was conducted. The results indicate that changes in HRV parameter values are positively correlated with changes in the severity of depression symptoms [40]. Celine K et al. [41] included 2250 depression patients and 1982 controls in a random effects meta-analysis. The results showed that depression patients had lower measurements of all HRVs than did healthy controls. This finding strengthens the evidence that a low HRV can potentially increase cardiovascular risk in depressed patients. Norie and colleagues also reported a correlation between depression and HRV [42]. This finding is in line with our findings. Hanife evaluated the function of the autonomic nervous system in patients with obsessive-compulsive disorder and healthy individuals by using HRV parameters. One study revealed that patients with obsessive-compulsive disorder had a significantly greater heart rate and high-frequency power (HF) of HRV than healthy individuals. Moreover, there were no significant differences in other HRV parameters between patients and healthy individuals [43]. Ying and colleagues' meta-analysis examined 4,897 anxiety patients and 5,559 control individuals and revealed a marked decrease in resting parasympathetic activity (HRV) among AD patients compared to that in the control group [44]. Reut Naim et al. investigated cardiovascular arousal (heart rate (HR) and heart rate variability (HRV)) in youths with irritable mood and reported a correlation between irritable mood and increased heart rate but decreased HRV [45].

Our results are inconsistent with most previous studies in terms of the association between anxiety and irritable mood and HRV. There are several possible reasons for this discrepancy. First, this could be explained by the fact that we employed 2 separate GWASs (one for anxiety or irritability and the other for HRV). The number of individuals with anxiety or irritability was rather low in the HRV GWAS. Another consideration is that the HRV GWAS data were not corrected for heart rate, which is strongly correlated with HRV [46]. Second, the occurrence of emotional disorders is caused by genetic and environmental factors. We investigated the relationship between emotional disorders and HRV from a genetic perspective. A lower HRV may be caused by systemic manifestations of anxiety or irritability, and the HRV is easily influenced by changes in blood pressure and physical activity. These confounding factors cannot be avoided in observational studies. MR studies can reduce confounding bias and suggest significant associations [47]. Finally, the MR study considered lifetime effects rather than short-term effects, which might explain the differences between our findings and previous literature [48]. Therefore, HRV testing is still necessary for patients with anxiety and irritable mood.

The link between affective activity and HRV has largely been attributed to dysregulation of the autonomic nervous system, resulting in a chronic shift toward increased sympathetic activity and decreased parasympathetic activity [49]. A higher HRV indicates enhanced control of

the parasympathetic nervous system. Elevated HRV is often considered a marker of psychological resilience and the capacity to react appropriately and flexibly to ever-changing environmental cues. In contrast, a decreased HRV suggests greater control of the sympathetic nervous system, which is often associated with increased impulsivity and greater difficulties in emotional regulation [50]. The neurovisceral integration model posits that shared neural networks support the effective regulation of emotions and HRV [51]. Vagally-mediated heart rate variability (vmHRV) represents a psychophysiological index of inhibitory control [52]. Emotional disorders are related to poor executive control of behavior, as the prefrontal cortex and parasympathetic nervous system are connected through the structure and function of the vagus nerve, and the prefrontal and limbic structures control HRV [52, 53]. Depression and other emotional disorders promote changes in the autonomic nervous system function of vagus nerve contraction and a decrease in prefrontal cortex activity, which are the main reasons for vagus nerve-mediated HRV reduction [54].

Our MR study has several strengths. First, in this MR analysis, the latest GWAS datasets were used to eliminate single-nucleotide polymorphisms associated with exposure and outcomes. Subsequently, we employed five different models to comprehensively investigate the potential relationship between emotional disorders and HRV, thus avoiding traditional confounding factors and inverse causality. Second, our analysis did not reveal any heterogeneity or directional pleiotropy, and sensitivity testing reinforced the stability and accuracy of causal findings. Finally, this study demonstrated a significant suggestive relationship from a genetic perspective, which supports the practical application of HRV detection. It is essential to acknowledge some of the limitations of this research. First, our MR technique employed data aggregated from a GWAS that included a European population. The absence of aggregated data from other regions limits its regional and racial effectiveness. Second, emotional disorders are influenced by subjective factors that are noticeable in research. Additionally, these factors can compromise the validity of the data statistics. Therefore, additional data should be collected and additional magnetic resonance imaging techniques applied to address confounding risk factors to obtain more stable and precise findings. In addition, sex was not differentiated in our study, and potential bias due to sex could not be excluded. Finally, additional research is needed to address the controversial findings of current observational studies.

## 5. Conclusion

The current two-sample MR study provides evidence for a negative correlation between depression and HRV and a positive correlation of irritable mood and anxiety with HRV. Severe depressive disorder, bipolar disorder, obsessive-compulsive disorder, manic disorder, and HRV were not found to be causal. These outcomes offer novel insights into the genetic aspects of the connection between emotional disorders and heart rate variability. Consequently, to safeguard cardiovascular well-being in people with emotional disorders, it is imperative to perform a clear analysis and monitor HRV.

## Supporting information

**S1 Table. Specific information on instrumental variables for seven types of emotional disorders.**
(DOCX)

**S2 Table. Heterogeneity and sensitivity analysis results for emotional disorders and HRV (pvRSA/HF).**
(DOCX)

**S3 Table. Heterogeneity and sensitivity analysis results for emotional disorders and HRV (RMSSD).**
(DOCX)

**S4 Table. Heterogeneity and sensitivity analysis results for emotional disorders and HRV (SDNN).**
(DOCX)

**S1 Fig. Scatter plot of heart rate variability (pvRSA/HF) and emotional disorders.** A. Depression.(broad) B. Major Depressive Disorder C. Obsessive Compulsive Disorder D. Bipolar Disorder E. Irritable Mood F. Anxiety Disorder G. Mania.
(DOCX)

**S2 Fig. Leave-one-out analysis of heart rate variability (pvRSA/HF) and emotional disorders.** A. Depression (broad) B. Major Depressive Disorder C. Obsessive Compulsive Disorder D. Bipolar Disorder. E. Irritable Mood F. Anxiety Disorder G. Mania.
(DOCX)

**S3 Fig. Scatter plot of heart rate variability (RMSSD) and emotional disorders.** A. Depression(broad) B. Major Depressive Disorder C. Obsessive Compulsive Disorder D. Bipolar Disorder E. Irritable Mood. F. Anxiety Disorder G. Mania.
(DOCX)

**S4 Fig. Leave-one-out analysis of heart rate variability traits (RMSSD) and emotional disorders.** A. Depression (broad). B. Major Depressive Disorder C. Obsessive Compulsive Disorder D. Bipolar Disorder. E. Irritable Mood F. Anxiety Disorder G. Mania.
(DOCX)

**S5 Fig. Scatter plot of heart rate variability (SDNN) and emotional disorders.** A. Depression(broad) B. Major Depressive Disorder C. Obsessive Compulsive Disorder D. Bipolar Disorder E. Irritable Mood F. Anxiety Disorder G. Mania.
(DOCX)

**S6 Fig. Leave-one-out analysis of heart rate variability (SDNN) and emotional disorders.** A. Depression (broad) B. Major Depressive Disorder C. Obsessive Compulsive Disorder D. Bipolar Disorder. E. Irritable Mood F. Anxiety Disorder G. Mania.
(DOCX)

## Author Contributions

**Conceptualization:** Wen Xie.

**Data curation:** Rui Wang.

**Methodology:** YunXiang Zhou.

**Writing – original draft:** Xu Luo.

**Writing – review & editing:** Xu Luo, Wen Xie.

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
