## [Decision Letter · Decision Letter 0]

5 Nov 2023

PONE-D-23-27049The relationship between emotional disorders and heart rate variability：A Mendelian randomizationPLOS ONE

Dear Dr. xie,

Thank you for submitting your manuscript to PLOS ONE. After careful consideration, we feel that it has merit but does not fully meet PLOS ONE’s publication criteria as it currently stands. Therefore, we invite you to submit a revised version of the manuscript that addresses the points raised during the review process.

We look forward to receiving your revised manuscript.

Kind regards,

Simone Battaglia

Guest Editor

PLOS ONE

5. PLOS requires an ORCID iD for the corresponding author in Editorial Manager on papers submitted after December 6th, 2016. Please ensure that you have an ORCID iD and that it is validated in Editorial Manager. To do this, go to ‘Update my Information’ (in the upper left-hand corner of the main menu), and click on the Fetch/Validate link next to the ORCID field. This will take you to the ORCID site and allow you to create a new iD or authenticate a pre-existing iD in Editorial Manager. Please see the following video for instructions on linking an ORCID iD to your Editorial Manager account: https://www.youtube.com/watch?v=_xcclfuvtxQ.

7. Please ensure that you refer to Figure 1 in your text as, if accepted, production will need this reference to link the reader to the figure.

Reviewers' comments:

Reviewer's Responses to Questions

**Comments to the Author**

1. Is the manuscript technically sound, and do the data support the conclusions?

Reviewer #1: Yes

Reviewer #2: Yes

2. Has the statistical analysis been performed appropriately and rigorously? 

Reviewer #1: Yes

Reviewer #2: Yes

3. Have the authors made all data underlying the findings in their manuscript fully available?

Reviewer #1: Yes

Reviewer #2: Yes

4. Is the manuscript presented in an intelligible fashion and written in standard English?

Reviewer #1: Yes

Reviewer #2: Yes

5. Review Comments to the Author

Reviewer #1: In the present study, entitled “The relationship between emotional disorders and heart rate variability: A Mendelian randomization”, authors utilize Mendelian randomization to study the development of heart rate variability (HRV) irregularities in people with various psychiatric disorders. To do so, they analysed public genome-wide association study datasets for emotional disorders and heart rate variability and by means of Mendelian randomization highlighted a causal relationship between depression, anxiety, and irritable mood on HRV irregularities.

Despite the interesting results found by the authors, there are some major limitations, affecting the clarity of the manuscript that the authors should address, as well as other relatively minor comments to improve the paper. In general, I think the idea of the study is interesting and might be of interest to the readers of PLOS One, but some adjustments are required before publication.

Major comments

- Throughout the manuscript the words used to define psychiatric disorders are incoherent between them. As an example, in the abstract depression is presented as “Generalized depression”, “Major depressive disorder”, and “Widespread depression”. These seem arbitrary definitions and I would suggest sticking to a single definition that is agreed upon by the scientific literature, such as “Major depressive disorder” if considering the example above.

- At the current state, the introduction prefaces how the study is carried out and the theoretical background, but it lacks a clear and direct statement about the scope of this work, which is only briefly mentioned in the abstract.

- In many circumstances the sentences used need a citation to support the content with evidence from the literature. This is evident especially, but not only, in the methods section. As an example, at page 3, line 71 the text reads: “Since mixed populations may lead to partial estimates, we restricted the genetic background of individuals included in the MR study to those of European ancestry”. I believe that such a sentence needs to be backed up by literature that supports this decision.

- Page 5, line 103 reads: “As EAF data could not be obtained for some samples, they were taken = 2/2 (Beta and Se data can be directly obtained).”. What does EAF stand for? It has not been used in any other part of the manuscript, nor has it been defined. What is “Se”? Again, it is not mentioned anywhere else. Moreover, the rationale behind this decision is not explained, nor it is supported by the bibliography (a problem mentioned in the comment above). I would suggest to carefully go over both the methods and results section of the manuscript to ensure that:

o All acronyms are appropriately defined and explained,

o Sentences that require a citation are given an appropriate one.

- Given the interdisciplinary readership of PLOS One, which comes from different academic backgrounds, a brief primer on Mendelian randomization in the introduction is highly recommended. On many occasions throughout the text the authors use words such as heterogeneity or horizontal pleiotropy which have a specific meaning in the context of MR but leave a non-expert reader wondering what they mean, as no explanation is given.

- It must be made clear why the author’s results provide causality. Suggested reads:

o Davey Smith, G., & Ebrahim, S. (2003). ‘Mendelian randomization’: can genetic epidemiology contribute to understanding environmental determinants of disease? International journal of epidemiology, 32(1), 1-22.

o Tian, D., Zhang, L., Zhuang, Z., Huang, T., & Fan, D. (2021). A two-sample Mendelian randomization analysis of heart rate variability and cerebral small vessel disease. Journal of clinical hypertension (Greenwich, Conn.), 23(8), 1608–1614. https://doi.org/10.1111/jch.14316

- Authors state throughout the manuscript about the causal relationship between mental health and HRV. But in these terms, it does not make sense. What is different in HRV about these patients as opposed to the healthy population? Based on the literature, I think it would be more appropriate to talk about a link between psychiatric disorders and a reduced HRV, hence a more regular heartbeat (see Koch et al., 2019 https://doi.org/10.1017/S0033291719001351; Chalmers et al., 2014 https://doi.org/10.3389/fpsyt.2014.00080). This needs to be stated clearly.

- Page 11, line 233: After stating multiple times that the results provide causality, the authors now state that their study failed to establish a causal relationship. Is this a typo? Moreover, immediately after it is suggested that additional (to what, since there is no imaging data in this study?) magnetic resonance imaging techniques are needed to address confounding factors, but I do not believe this is in any way relevant with the topic at hand. I suggest the authors carefully revise this section of the manuscript.

- Page 12, line 241: The last sentence needs to be rephrased. “to safeguard cardiovascular well-being,” seems to refer to everyone, but in this case it would be better to specify the “cardiovascular well-being of people with emotional disorders”. Moreover, the authors suggest a clear analysis, but not what needs to undergo such analysis. Notwithstanding, I would not say that this is imperative, but it may be a suggestion for future research and clinical practice. Finally, since the correlation between emotional disorders and aberrant heart rate behavior is a well-established phenomenon (Battaglia et al., 2023a https://10.1016/j.neubiorev.2023.105163; Battaglia et al., 2023b https://10.1111/acps.13602), I would put more emphasis on the idea of causality that emerges from the results and why it is relevant.

Minor comments

- I believe the abstract goes too much into detail in regard to both the methodology and the results section. I suggest removing the last sentence in the “Methods” subsection, as well as removing the result of the analyses in parentheses and simply describing the findings in more direct and concise sentences.

- The manuscript would benefit from a native speaker proofreading service, as there are some grammatical errors throughout the text.

- Page 9, line 178 and 180: Change SNDD to SDNN

Reviewer #2: Thank you for the opportunity to review this interesting manuscript. The manuscript examines the causative associations between emotional disorders and heart rate variability (HRV) measures using Mendelian randomization. Five models were included for the MR analysis. The models show causal relationships between widespread depression, angry emotions and one HRV trait (PVRSA/HF), and between the HRV trait (SDNN) and angry emotions. Finally, Anxiety and HRV trait (RMSSD) have a causal relationship. The manuscript is well written, methodology and sample size are appropriate. I have few comments.

Introduction

Pag 4. Previous studies (if any) linking specific HRV measures and emotional disorders should be reported to clarify present study predictions and hypotheses.

Methods

Although the methods and the statistical strategy are clearly defined, I suggest to include a schematic figure that synthetize the main statistical procedures within the MR (IV, exposure, model application, outcomes etc) (Tian D, Zhang L, Zhuang Z, Huang T, Fan D. A two-sample Mendelian randomization analysis of heart rate variability and cerebral small vessel disease. J Clin Hypertens (Greenwich). 2021 Aug;23(8):1608-1614. doi: 10.1111/jch.14316)

The selection of instrumental variables is based on different analytic steps and parameters, I suggest to include appropriate references that justify the used parameters.

Discussion

Results show that different emotional diseases are linked to specific HRV indices. May the authors discuss the possible functional mechanisms that can explain these differential effects? (S. Battaglia, C. Nazzi, J.F. Thayer, Fear-induced bradycardia in mental disorders: Foundations, current advances, future perspectives, Neuroscience & Biobehavioral Reviews, Volume 149,2023,105163,ISSN 0149-7634 https://doi.org/10.1016/j.neubiorev.2023.105163; Jung W, Jang KI, Lee SH. Heart and Brain Interaction of Psychiatric Illness: A Review Focused on Heart Rate Variability, Cognitive Function, and Quantitative Electroencephalography. Clin Psychopharmacol Neurosci. 2019 Nov 20;17(4):459-474. doi: 10.9758/cpn.2019.17.4.459)

Minor:

ABSTRACT: widespread depression and HRVT (PVRSA/HF). The existence 25 of a causal relationship

26 between HRVT (pvRSA/HF). Use the same format for pvRSA/HF

Keywords are a simple repetition of the title, I suggest to use different MeSH words

6. PLOS authors have the option to publish the peer review history of their article (what does this mean?). If published, this will include your full peer review and any attached files.

Reviewer #1: No

Reviewer #2: No

---

## [Author Response · Author response to Decision Letter 0]

19 Dec 2023

Reviewer 1

Thank you very much for your valuable comments to improve the quality of our manuscript. All questions have been answered in itemized descriptions as shown below.

Major comments

1.Throughout the manuscript the words used to define psychiatric disorders are incoherent between them. As an example, in the abstract depression is presented as “Generalized depression”, “Major depressive disorder”, and “Widespread depression”. These seem arbitrary definitions and I would suggest sticking to a single definition that is agreed upon by the scientific literature, such as “Major depressive disorder” if considering the example above.

Response: 

Many thanks for your valuable comments for improving the quality of our manuscript. Accordingly, Depression (broad) was revised to broad depression in the Revised Manuscript with Track Changes and Manuscript. (References[19]).

2.At the current state, the introduction prefaces how the study is carried out and the theoretical background, but it lacks a clear and direct statement about the scope of this work, which is only briefly mentioned in the abstract.

Response:

Many thanks for your valuable comments for improving the quality of our manuscript. Accordingly, the correction has been made in the Revised Manuscript with Track Changes (Page: 4, line: 73-76) and Manuscript(Page: 3, line: 56-58).

3.In many circumstances the sentences used need a citation to support the content with evidence from the literature. This is evident especially, but not only, in the methods section. As an example, at Page: 3, line 71 the text reads: “Since mixed populations may lead to partial estimates, we restricted the genetic background of individuals included in the MR study to those of European ancestry”. I believe that such a sentence needs to be backed up by literature that supports this decision.

Response:

Many thanks for your valuable comments for improving the quality of our manuscript. Accordingly, the correction has been made in the Revised Manuscript with Track Changes and Manuscript (References[18]).

4.Page: 5, line 103 reads: “As EAF data could not be obtained for some samples, they were taken = 2/2 (Beta and Se data can be directly obtained).”. What does EAF stand for? It has not been used in any other part of the manuscript, nor has it been defined. What is “Se”? Again, it is not mentioned anywhere else. Moreover, the rationale behind this decision is not explained, nor it is supported by the bibliography (a problem mentioned in the comment above). I would suggest to carefully go over both the methods and results section of the manuscript to ensure that:

o All acronyms are appropriately defined and explained,

o Sentences that require a citation are given an appropriate one.

Response:

Many thanks for your valuable comments for improving the quality of our manuscript. Accordingly, the correction has been made in the Revised Manuscript with Track Changes (Page:6, lines:123-127)and Manuscript (Page:5, lines:100-104).

5.Given the interdisciplinary readership of PLOS One, which comes from different academic backgrounds, a brief primer on Mendelian randomization in the introduction is highly recommended. On many occasions throughout the text the authors use words such as heterogeneity or horizontal pleiotropy which have a specific meaning in the context of MR but leave a non-expert reader wondering what they mean, as no explanation is given.

Response:

Many thanks for your valuable comments for improving the quality of our manuscript. Accordingly, the correction has been made in the Revised Manuscript with Track Changes (Page:7, lines:149-151,lines:155-157) and Manuscript (Page:6, lines:125-127, lines:131-133).

6.It must be made clear why the author’s results provide causality. Suggested reads:

o Davey Smith, G., & Ebrahim, S. (2003). ‘Mendelian randomization’: can genetic epidemiology contribute to understanding environmental determinants of disease? International journal of epidemiology, 32(1), 1-22.

o Tian, D., Zhang, L., Zhuang, Z., Huang, T., & Fan, D. (2021). A two-sample Mendelian randomization analysis of heart rate variability and cerebral small vessel disease. Journal of clinical hypertension (Greenwich, Conn.), 23(8), 1608–1614. https://doi.org/10.1111/jch.14316

Response:

Many thanks for your valuable comments for improving the quality of our manuscript. Accordingly, the correction has been made in the Revised Manuscript with Track Changes (Page:8, lines:166-169; Page:9, lines:180-182;Page:10, lines:191-193;Page:13-14, lines:247-268) and Manuscript (Page:7, lines:141-144; Page:8 lines:154-155; Page:9 lines:165-166; Page:12, lines:217-230).

7.Authors state throughout the manuscript about the causal relationship between mental health and HRV. But in these terms, it does not make sense. What is different in HRV about these patients as opposed to the healthy population? Based on the literature, I think it would be more appropriate to talk about a link between psychiatric disorders and a reduced HRV, hence a more regular heartbeat 

(see Koch et al., 2019 https://doi.org/10.1017/S0033291719001351;

Chalmers etal.2014 https://doi.org/10.3389/fpsyt.2014.00080). This needs to be stated clearly.

Response:

Many thanks for your valuable comments for improving the quality of our manuscript. Accordingly, the correction has been made in the Revised Manuscript with Track Changes (Page:1, lines:10; Page: 3, lines:62-64; Page:11, lines:207-210) and Manuscript (Page:1, lines:8-9; Page:3 lines:48-50; Page:10 lines:179-182). 2. MR analysis showed that genetic susceptibility to broad depression was negatively correlated with HRV. However,anxiety genetic susceptibility to irritability emotions was positively correlated with HRV. Therefore, this study systematically investigated the potential link between genetic determined emotional disorders and HRV. The possible reasons for the discrepancy between the results of this study and clinical research have been added in the Revised Manuscript with Track Changes ( Page:12-13, lines:232-246) and Manuscript (Page:11-12, lines:204-216).

8.Page: 11, line 233: After stating multiple times that the results provide causality, the authors now state that their study failed to establish a causal relationship. Is this a typo? Moreover, immediately after it is suggested that additional (to what, since there is no imaging data in this study?) magnetic resonance imaging techniques are needed to address confounding factors, but I do not believe this is in any way relevant with the topic at hand. I suggest the authors carefully revise this section of the manuscript. Response:

Many thanks for your valuable comments for improving the quality of our manuscript. Accordingly, the correction has been made in the Revised Manuscript with Track Changes (Page:15, lines:287-289).

9.Page: 12, line 241: The last sentence needs to be rephrased. “to safeguard cardiovascular well-being,” seems to refer to everyone, but in this case it would be better to specify the “cardiovascular well-being of people with emotional disorders”. Moreover, the authors suggest a clear analysis, but not what needs to undergo such analysis. Notwithstanding, I would not say that this is imperative, but it may be a suggestion for future research and clinical practice. Finally, since the correlation between emotional disorders and aberrant heart rate behavior is a well-established phenomenon (Battaglia et al., 2023a https://10.1016/j.neubiorev.2023.105163; Battaglia et al., 2023b https://10.1111/acps.13602), I would put more emphasis on the idea of causality that emerges from the results and why it is relevant.

Response:

Many thanks for your valuable comments for improving the quality of our manuscript. Accordingly, the correction has been made in the Revised Manuscript with Track Changes (Page:15 lines:296-298, lines:301-303)and Manuscript (Page:13, lines:247-252).

Minor comments

1.I believe the abstract goes too much into detail in regard to both the methodology and the results section. I suggest removing the last sentence in the “Methods” subsection, as well as removing the result of the analyses in parentheses and simply describing the findings in more direct and concise sentences.

Response:

Many thanks for your valuable comments for improving the quality of our manuscript. Accordingly, the correction has been made in the Revised Manuscript with Track Changes(Page:1-2,line:16-35)and Manuscript (Page:1-2, lines:12-24).

2.The manuscript would benefit from a native speaker proofreading service, as there are some grammatical errors throughout the text.

Response:

Many thanks for your valuable comments for improving the quality of our manuscript. Accordingly, the correction has been made in the Revised Manuscript with Track Changes and Manuscript.

3. Page: 9, line 178 and 180: Change SNDD to SDNN

Response:

Many thanks for your valuable comments for improving the quality of our manuscript. Accordingly, the correction has been made in the Revised Manuscript with Track Changes(Page:11,line:206) and Manuscript (Page:10, lines:179).

Reviewer 2: Thank you for the opportunity to review this interesting manuscript. The manuscript examines the causative associations between emotional disorders and heart rate variability (HRV) measures using Mendelian randomization. Five models were included for the MR analysis. The models show causal relationships between widespread depression, angry emotions and one HRV trait (PVRSA/HF), and between the HRV trait (SDNN) and angry emotions. Finally, Anxiety and HRV trait (RMSSD) have a causal relationship. The manuscript is well written, methodology and sample size are appropriate. I have few comments.

Response:Thank you very much for your valuable comments to improve the quality of our manuscript. All questions have been answered in itemized descriptions as shown below.

1. Introduction. Pag 4. Previous studies (if any) linking specific HRV measures and emotional disorders should be reported to clarify present study predictions and hypotheses.

Response:

Many thanks for your valuable comments for improving the quality of our manuscript. Accordingly, the correction has been made in the Revised Manuscript with Track Changes (Page:3,line:62-64) and Manuscript (Page:3, lines:48-50).

2.Methods. Although the methods and the statistical strategy are clearly defined, I suggest to include a schematic figure that synthetize the main statistical procedures within the MR (IV, exposure, model application, outcomes etc) (Tian D, Zhang L, Zhuang Z, Huang T, Fan D. A two-sample Mendelian randomization analysis of heart rate variability and cerebral small vessel disease. J Clin Hypertens (Greenwich). 2021 Aug;23(8):1608-1614. doi: 10.1111/jch.14316)

Response:

Many thanks for your valuable comments for improving the quality of our manuscript. Accordingly, the correction has been made in the Revised Manuscript with Track Changes and Manuscript( Figure 1).

3.The selection of instrumental variables is based on different analytic steps and parameters, I suggest to include appropriate references that justify the used parameters.

Response:

Many thanks for your valuable comments for improving the quality of our manuscript. Accordingly, the correction has been made in the Revised Manuscript with Track Changes and Manuscript (References:26-30)

4.Discussion.Results show that different emotional diseases are linked to specific HRV indices. May the authors discuss the possible functional mechanisms that can explain these differential effects? (S. Battaglia, C. Nazzi, J.F. Thayer, Fear-induced bradycardia in mental disorders: Foundations, current advances, future perspectives, Neuroscience & Biobehavioral Reviews, Volume 149,2023,105163,ISSN 0149-7634 https://doi.org/10.1016/j.neubiorev.2023.105163; Jung W, Jang KI, Lee SH. Heart and Brain Interaction of Psychiatric Illness: A Review Focused on Heart Rate Variability, Cognitive Function, and Quantitative Electroencephalography. Clin Psychopharmacol Neurosci. 2019 Nov 20;17(4):459-474. Doi: 10.9758/cpn.2019.17.4.459)

Response:

Many thanks for your valuable comments for improving the quality of our manuscript. Accordingly, the correction has been made in the Revised Manuscript with Track Changes. (Page:12-14,line:232-268)and Manuscript (Page:11-12, lines:204-230).

Minor:

1.ABSTRACT: widespread depression and HRVT (PVRSA/HF). The existence 25 of a causal relationship

26 between HRVT (pvRSA/HF). Use the same format for pvRSA/HF

Response:

Many thanks for your valuable comments for improving the quality of our manuscript. Accordingly, the correction has been made in the Revised Manuscript with Track Changes and Manuscript. 

2.Keywords are a simple repetition of the title, I suggest to use different MeSH words

Response:

Many thanks for your valuable comments for improving the quality of our manuscript. Accordingly, the correction has been made in the Revised Manuscript with Track Changes (Page:2,line:41-42)and Manuscript (Page:2, lines:27-28).

---

## [Decision Letter · Decision Letter 1]

18 Jan 2024

PONE-D-23-27049R1The relationship between emotional disorders and heart rate variability：A Mendelian randomizationPLOS ONE

Dear Dr. xie,

Thank you for submitting your manuscript to PLOS ONE. After careful consideration, we feel that it has merit but does not fully meet PLOS ONE’s publication criteria as it currently stands. Therefore, we invite you to submit a revised version of the manuscript that addresses the points raised during the review process.

We look forward to receiving your revised manuscript.

Kind regards,

Simone Battaglia

Guest Editor

PLOS ONE

Journal Requirements:

Reviewers' comments:

Reviewer's Responses to Questions

**Comments to the Author**

1. If the authors have adequately addressed your comments raised in a previous round of review and you feel that this manuscript is now acceptable for publication, you may indicate that here to bypass the “Comments to the Author” section, enter your conflict of interest statement in the “Confidential to Editor” section, and submit your "Accept" recommendation.

Reviewer #1: (No Response)

Reviewer #2: All comments have been addressed

2. Is the manuscript technically sound, and do the data support the conclusions?

Reviewer #1: Yes

Reviewer #2: Yes

3. Has the statistical analysis been performed appropriately and rigorously? 

Reviewer #1: Yes

Reviewer #2: Yes

4. Have the authors made all data underlying the findings in their manuscript fully available?

Reviewer #1: Yes

Reviewer #2: Yes

5. Is the manuscript presented in an intelligible fashion and written in standard English?

Reviewer #1: No

Reviewer #2: Yes

6. Review Comments to the Author

Reviewer #1: All comments have been addressed, but there are still many typos, grammatical errors and punctuation errors. Despite the authors stated that the manuscript already underwent proofreading, I strongly suggest having it edited once more, referring to a professional native speaker proofreader. The inaccuracies present within the manuscript as of now negatively impact on its quality, and need to be addressed before publication.

Reviewer #2: I thank the authors for their response. I believe they addressed all of my comments satisfactorily. I have no additional comments to add.

7. PLOS authors have the option to publish the peer review history of their article (what does this mean?). If published, this will include your full peer review and any attached files.

Reviewer #1: No

Reviewer #2: No

---

## [Author Response · Author response to Decision Letter 1]

25 Jan 2024

Reviewer 1

Thank you very much again for your valuable comments to improve the quality of our manuscript. All questions have been answered in itemized descriptions as shown below. 

Major comments

1.All comments have been addressed, but there are still many typos, grammatical errors and punctuation errors. Despite the authors stated that the manuscript already underwent proofreading, I strongly suggest having it edited once more, referring to a professional native speaker proofreader. The inaccuracies present within the manuscript as of now negatively impact on its quality, and need to be addressed before publication.

Response: 

Many thanks for your valuable comments for improving the quality of our manuscript. Accordingly, the correction has been made in the Revised Manuscript with Track Changes and Manuscript.

Reviewer 2: I thank the authors for their response. I believe they addressed all of my comments satisfactorily. I have no additional comments to add.

Thank you very much again for your valuable comments to improve the quality of our manuscript.

---

## [Decision Letter · Decision Letter 2]

5 Feb 2024

The relationship between emotional disorders and heart rate variability: A Mendelian randomization study

PONE-D-23-27049R2

Dear Dr. Xu Luo,

We’re pleased to inform you that your manuscript has been judged scientifically suitable for publication and will be formally accepted for publication once it meets all outstanding technical requirements.

Kind regards,

Simone Battaglia

Guest Editor

PLOS ONE

Additional Editor Comments (optional):

Reviewers' comments:

Reviewer's Responses to Questions

**Comments to the Author**

1. If the authors have adequately addressed your comments raised in a previous round of review and you feel that this manuscript is now acceptable for publication, you may indicate that here to bypass the “Comments to the Author” section, enter your conflict of interest statement in the “Confidential to Editor” section, and submit your "Accept" recommendation.

Reviewer #1: All comments have been addressed

2. Is the manuscript technically sound, and do the data support the conclusions?

Reviewer #1: Yes

3. Has the statistical analysis been performed appropriately and rigorously? 

Reviewer #1: Yes

4. Have the authors made all data underlying the findings in their manuscript fully available?

Reviewer #1: Yes

5. Is the manuscript presented in an intelligible fashion and written in standard English?

Reviewer #1: Yes

6. Review Comments to the Author

Reviewer #1: (No Response)

7. PLOS authors have the option to publish the peer review history of their article (what does this mean?). If published, this will include your full peer review and any attached files.

Reviewer #1: No

---

## [Editor Report · Acceptance letter]

26 Feb 2024

PONE-D-23-27049R2 

PLOS ONE

Dear Dr. Xie, 

I'm pleased to inform you that your manuscript has been deemed suitable for publication in PLOS ONE. Congratulations! Your manuscript is now being handed over to our production team.

Kind regards, 

on behalf of

Dr. Simone Battaglia 

Guest Editor

PLOS ONE